# On the Way to Eco-Innovations in Agriculture: Concepts, Implementation and Effects at National and Local Level. The Case of Poland

## Michał Dudek [1,*] and Wioletta Wrzaszcz [2]

1    Department of European Integration, Institute of Rural and Agricultural Development, Polish Academy of Sciences, 72 Nowy Świat St., 00-330 Warsaw, Poland
2    Department of Agricultural Economics, Agricultural Policy and Rural Development, Institute of Agricultural and Food Economics-National Research Institute, 20 Świętokrzyska St., 00-002 Warsaw, Poland; wioletta.wrzaszcz@ierigz.waw.pl
*    Correspondence: mdudek@irwirpan.waw.pl

**Abstract:** The aim of the study was to provide the examples of eco-innovations in agriculture relating to the concept of sustainable development and the indication of their conditions. Quantitative and qualitative methods were applied to the research, namely: descriptive statistical and economic analysis of the Polish Farm Accountancy Data Network (FADN) data and Statistics Poland data, as well as case studies of organic food producers, covering the years 2005–2019. Indicated information sources, encompassing long time span of analysis and various data collections, allowed presenting the complementary picture of eco-innovations at the sector and farm levels. The research examined the different types of ecological innovations in Polish agriculture, including: (1) organisational innovations with an institutional background (e.g., the organic farming support and greening mechanism of the Common Agricultural Policy (CAP)—implemented in the family farming sector); and (2) the product, marketing, process and organisational innovations in selected organic farms that were individual farmers' initiatives. On the one hand, the research documented the effectiveness of new agricultural policy solutions in the agricultural sector that are examples of organisational eco-innovations. During 2005–2016, the certification system, as well as policy support, contributed to the development of organic farms in Poland in terms of the growth in the share of this type of holdings in total (from 0.5% to 4.6%) and in the overall utilised agricultural area (UAA) (from 0.3% to 3.7%). Moreover, during 2014–2015, as a result of the greening in agricultural holdings, the area sown with pulses and papilionaceous, i.e., crops improving soil structure and protecting soils, rose by 174% and 161%, respectively. On the other hand, the case studies conducted showed that the food producers' knowledge and skills combined with a favourable local economic and social situation, as well as institutional support, played a key role in the process of the emergence of eco-innovations. Among those factors, the respondents' individual characteristics associated with attitudes towards farming and the social, human and physical capital passed on by family members should be highlighted. This paper contributes to existing literature in two ways. First, this study combines both quantitative and qualitative (including in-depth interviews) approaches to eco-innovations at the micro and macro level of analysis. Second, by differentiating two approaches to ecological innovations, namely the conventional and the sustainable, the article indicates and considers the key factors favourable to the latter.

**Keywords:** eco-innovation; sustainable development of agriculture; organic farming; greening; the CAP

## 1. Introduction

In the scientific literature, public policy and economic practice, increasing attention is being paid to ecological innovations, also known as eco-innovations [1–4]. The growing interest in new solutions that are beneficial to nature and the climate, conditioned by technical progress and changes in market participant attitudes, covers different sectors of the economy, including agriculture. Agriculture has a significant impact on the state of the natural environment and the pace and direction of climate change, as well as on consumer health, which determines the need to look for new "eco" solutions [5]. However, eco-innovation poses a challenge to the scientific community, as there is also a lack of in-depth empirical studies and qualitative analyses, including in the area of agriculture [6].

At the same time, despite growing attention to environmental projects, the predominant industrial agricultural production model is based on new technological solutions whose social and environmental character has not been sufficiently recognised [7,8]. This situation is essentially based on the paradigm of growth, highlighting the need to increase production efficiency, while possibly reducing the negative impact on the natural environment [9]. Eco-innovation in agriculture that relates to the concept of sustainable development presents another order of priorities. In this case, environmental boundaries define the business framework, in line with the assumptions of ecological economics [10]. A social aspect that highlights the need for the primacy of consumer health, income parity in agriculture and the need to maintain the viability of rural areas also plays an important role [10,11].

New and environmental solutions are expected in agriculture in Poland. The environmental awareness of society, as well as the demand for healthy food, is growing. Despite these changes, the deterioration of the state of natural environment and negative effects of climate change are increasingly observed [12]. The CAP is a policy for all European Union (EU) member states which is managed and funded at the supranational level. This policy covers several market and non-market regulations aimed at production, environmental and territorial issues [13]. An important role for rural and agricultural areas is attributed also to another EU policy—the cohesion policy—which supports the socio-economic development of these regions.

EU policy instruments, including the certification systems or greening mechanism, may contribute to the popularisation of eco-innovation. The latter is the effect of the 2013 direct-payment system modification, aimed at increasing the environmental friendliness of agricultural production and counteracting climate change. Another example of institutional action towards eco-innovation implementation is support for organic farming. This is a rapidly growing segment of the world's agri-food economy and a source of new, cost-effective products and services that create positive externalities for the natural environment and society [14]. Organic farming is an example of sustainable agriculture form that contributes to the conservation of biodiversity and the protection of natural resources, as well as healthy food production [15–17]. The essence of organic farming comes down to a holistic approach to management, which takes into account, on the one hand nature processes and on the other hand ethical values [18].

The studies carried out thus far have mainly focused on diagnosing the level of ecological innovation in different countries and sectors through indicator methods and surveys of selected groups of enterprises [19,20]. The studies have rarely addressed the conditions and effects of eco-innovation or their local dimension. One of the valuable ways of analysing complex and ambiguous phenomena such as eco-innovation is the case-study type of research [7,21].

The aim of this study was to provide examples of eco-innovations in agriculture relating to the concept of sustainable development and the indication of their conditions. Using the classic OECD innovation perspective, the article considers different types of eco-innovation, including the product, marketing, process and organisational ones [22,23]. The research was carried out at the sectoral level (the scale of implementation of the eco-innovations in the family-farm population in Poland) and local one (by carrying out an in-depth case study of organic food producers).

The paper begins with the presentation of the conventional and then the sustainable approach to the creation of ecological innovation in agriculture. This part also highlights the conditions of

eco-innovations in agriculture at various different levels, namely: local (food producer perspective) and sectoral ones. Secondly, Section 3 describes the methods used in the study to analyse eco-innovation in agriculture and characterises data sources. Following this, the research results in the scope of eco-innovation conditions and the effects of their implementation in the Polish agricultural sector are presented. The system of organic farming support and greening mechanism is considered regarding organisational eco-innovations. Next, a detailed illustration of selected case studies, i.e., organic production farms, is presented. The Section 5 focuses on interpretation of the results obtained and juxtaposing them with other views of the problem examined in the literature on the subject. Lastly, the paper ends with key conclusions.

## 2. Theoretical Background: The Conventional vs. a Sustainable Approach to Ecological Innovations in Agriculture

### 2.1. Conventional Approach

Nowadays, the most common understanding of eco-innovations is derived from a popular, conventional approach to innovation presented in mainstream economics, which can be put together with the industrial agriculture paradigm [23,24]. Within the conventional approach, an imperative of production growth forms the main premise for the introduction on ecological innovations. Subsequently, this process is also driven by the need to reduce pressure on the environment and climate, motivated by the green objectives of public institutions and health-promoting market trends [9,25]. On the one hand, the need to implement eco-innovation is enforced by the increasing world population and growing food demand and, on the other hand, by efforts to mitigate the negative environmental externalities [26].

In doing so, to achieve its economic and environmental goals, eco-innovation offers a unique solution which simultaneously guarantees business profitability. Such an approach implies a focus on technological and capital-intensive changes that make a reduction in the use of natural resources and limitation of negative externalities possible while maximising profit.

The technological and profit-maximising requirement of innovation is strictly linked with the need to involve modern management techniques and high-skilled employees, which primarily makes the corporations and large agri-food sector companies the perfect place for such projects [27]. The novelty of the products and services created is usually based on a reconciliation of the precedence of the business entity's economic interest with its social responsibility. The effects of biological progress (introduction of new plants varieties and breeding animal species), biotechnology and nanotechnology (plant and genomic selection, in vitro animal cells culture, nanosensors and nanoparticles), as well as the Internet of Things, robotisation, automation, blockchain and various digital technologies (e.g., big data, specialised software for agricultural production management, cloud computing, artificial intelligence, precision farming, drones and sensors), are often invoked as examples of such eco-innovations [28–30]. The above-mentioned solutions are disruptive, because they bring about market changes. At the same time, the conventionally interpreted agricultural eco-innovations contribute to the reduction of greenhouse-gas emissions and the limitation of the use of water, medicines, pesticides and mineral fertilisers, as well as facilitating higher unit productivity in livestock and crop production [28,31].

Due to a lack of sufficient data on ecological innovations and problems with measurement and conceptualisation of this phenomenon, the conventionally oriented research on their environmental and economic effects for farmers and consumers has so far rarely been studied [32–34]. The traditional analyses of eco-innovations have been multi-sectoral and related to large data bases. Their objectives were often to measure the potential and state of eco-innovativeness using indices for sectors, economies and regions. The components of the level of innovativeness assessment were indirect and inter alia covered patent information, results of material flows, sizes of input, output and consumption or the productivity of resource consumption [23,35].

Seeking the causes of new product and organisational solutions in the conventional approach, the researchers focused on the impact of various exogenous conditions, particularly socio-economic and organisational [35,36]. The interest in the latter concentrated on legal, fiscal and social support [2,24].

In such studies, an important role, conducive to eco-innovativeness, was attributed to endogenous factors which were directly linked with economic entities. In this context, in the case of farmers, the influence of several socio-demographic and socio-psychological variables—namely age, sex, the level of education and skills and the willingness to take risks—were analysed [4,31]. In line with this, various farm characteristics were also taken into consideration (e.g., economic potential and size, land property and ICT usage) [32]. The other dimension of ecological innovation—the mezzo level conditions—were described by an analysis of the situation on the local market where an economic entity operated. In this case, the level of demand and the structure and characteristics of a local food chain were of particular importance for the process of creating ecological innovation in agriculture [7].

*2.2. Sustainable Approach*

Unlike conventional theory and eco-innovation practice, there is a research trend that relates to the sustainability paradigm. This combines with a change in the perception of innovation and the specificity of its creation, which has been shaped by criticism of the functioning of industrial agri-food systems [37]. The significance of eco-innovations in agriculture is particularly important because it emphasises the close dependence of agriculture on natural conditions and resources, including the state of the soil and water and the provision of ecosystem services. In addition to the environmental aspect, the implementation of sustainability principles takes the economic and social aspects into account [14,38]. This means that, in addition to respect for natural resources, projects in line with this concept should take account of the implementation of the population's needs related to the delivery of food, income generation and fair access to economic and natural resources. In this perspective, eco-innovation should be seen as viable, cost-effective and beneficial to the natural environment [25,39].

The paradigm of sustainable agriculture assumes that it is possible to reconcile environmental, social and economic issues, given the integration of production conditions and resources embedded locally as environmental innovation factors [40]. The farmers' knowledge of agricultural production and their immediate farm surroundings, enabling such ecosystem management to be beneficial for the environment, their own economic interests and the standard of living of the local community, plays a decisive role [37]. Thus, a reserve of innovative projects may concern a potentially large group of agricultural producers with differing knowledge, capacities, experience, ways of farm management or social and environmental conditions of farming.

Initiative creation, responding to the needs of a particular producer or community, should generally be combined not only with the use of general knowledge acquired in educational or training institutions, but also with specific knowledge from different fields, based on individual agricultural practice (experience and experimentation), knowledge of local resources and established social relationships [37]. In the creation of ecological innovations in agriculture, the individual capacities, labour inputs, entrepreneurship and the ability to combine and organise various sources of information and available resources, as well as openness and willingness to cooperate, therefore play a major role [2,28].

In view of the ambiguity, subjectivity and insufficient reinforcement of eco-innovation theory, the need for an in-depth analysis based on context and the location of the conditions and forms of implementation is evident. An example of such research was the study of the mechanisation of agriculture in Nepal or an analysis of the case of Taiwanese small farmers combining organic coffee production with tourism and educational services [41,42]. The specificities of eco-innovation translate, firstly, into the analysis of a number of explanatory factors at different levels of analysis. Secondly, the diverse background and circumstances of the generation of these phenomena suggest the need for a different research methodology. Methodological pluralism [24,43] is created by different scientific techniques, including direct interviews, case studies, observations or traditional surveys [37,44]. Interdisciplinary research has a practical advantage and can contribute to deepening and broadening the theory and practice of eco-innovation.

In addition to grass-roots and local conditions conducive to the development of ecological innovation in agriculture, which is part of the sustainability paradigm, the institutional factor in the form of agricultural policy plays a significant role. For a long time, increasing agricultural production and agricultural land and labour productivity has been the primary objective of agricultural research and innovation in the (EU). One of the main reasons for government intervention in innovation is the fact that, due to market failures, the level of innovation has been sub-optimal [45]. Since 1992, the next CAP reforms have been applied in a series of steps to achieve environmental progress in the agricultural sector. In 2013 the CAP reform of direct payments, a greening mechanism, was established and played a role in the popularisation of eco-innovation in agriculture [46]. Greening requirements can also have a positive impact on innovation, notably by favouring the development of agricultural practices and systems that are more sustainable from an environmental point of view [47]. One of the main organisational forms of sustainable agriculture particularly predisposed to the development of eco-innovation is organic farming. This is a given set of farm practices that emphasise ecological sustainability [48]. Organic-farming systems activate natural production mechanisms through the use of environmentally friendly production conducive to soil fertility, animal health and healthy food products [15]. This system of agricultural production can bring different environmental effects, but most of them are more beneficial in comparison to conventional production [49]. Organic agriculture stresses the need to use farmer and farming-community knowledge, particularly in the scope of farm organisation to create a reservoir of adaptations [50]. The most important practices on the level of organic farm concern crop rotation, natural-resource management, appropriate selection of seeds and breeds, sustainable fertilisation and plant protection planning and the use of innovative and low-budget techniques [48]. This management system is also supported by the CAP. The post-2020 CAP will introduce a more ambitious purpose by implementing green architecture—the European Green Deal strategy—which will directly and indirectly support eco-innovations [51,52].

## 3. Materials and Methods

Quantitative and qualitative methods were applied to the research, namely: descriptive statistical and economic analysis of the Polish FADN data and Statistics Poland data, as well as case studies of organic food producers covering the years 2005–2019. Indicated information sources, encompassing long time span of analysis and various data collections, allowed presenting complementary picture of eco-innovations at the sector and the farms' level. The research examined the different types of ecological innovations in Polish agriculture, including: (1) organisational innovations with an institutional background—the organic farming support and greening mechanism of the CAP; and (2) the product, marketing, process and organisational innovations in selected organic farms.

### 3.1. Data Sources and Methods: Organic Farming Development in Poland

The first empirical case of organisational eco-innovation implementation in Poland was an analysis of certified organic farms. The organic-farm research was based on public data of Statistics Poland—the 2005 and 2016 Farm Structure Survey (FSS) results. These data were collected on the basis of a uniform methodology that made it possible to investigate trends Polish agriculture, including organic farming. The analysis concerned all individual agricultural holdings with at least 1 ha of agricultural land maintained in good agricultural and environmental condition. The research focused on individual organic farms (the number of organic farms with a production certificate). The farms' descriptive statistical analysis concerned their production and economic potential. The results were illustrated against the background of all individual farms in Poland, making it possible to identify the ranges of convergence and diversity in the development of farms in total compared to organic ones [53].

### 3.2. Data Sources and Methods: Implementation of Greening in Agriculture in Poland

The second empirical case of organisational eco-innovation implementation in Poland was an analysis of the greening mechanism. Descriptive statistical analysis was used to present the scale

of changes after greening implementation. Greening practices, implemented in 2015, are specified in the European Commission (EC) regulations that indicate the importance of crop diversification in the context of soil-quality improvement; the maintenance of permanent grasslands to ensure carbon sequestration, soil protection and biodiversity; and the maintenance of ecological focus areas that guarantee biodiversity at farm level [54,55]. Depending on the area of arable land used and the share of permanent grassland, farmers are required to follow one, two or three greening practices. These include: (a) diversification of crops (applicable to farms with an arable area of 10 ha or more); (b) maintenance of Ecological Focus Areas (EFA) on at least 5% of arable land (applies to farms with an arable area of 15 ha or more); and (c) maintenance of permanent grassland (the ratio of grassland to total agricultural area may not decrease by more than 5% compared to the reference ratio) [56].

The above formal requirements were the point of reference in the research methodology, including selection of farms and appropriate indicators to evaluate organisational eco-innovation implementation. The paper is based on a panel of 5700 individual farms in the Polish FADN. These farms kept agricultural accounting in both 2014 and 2015 (before and after greening implementation). The study does not cover farms exempt from the greening requirement (the smallest farms—below 10 ha of arable land) and those that apply equivalent practices to greening ones. The farm panel analysed was divided into two groups, i.e., farms with 10–15 ha of arable land (smaller farms obliged to crop diversification) and farms with of 15 ha of arable land and over (larger farms obliged to meet EFA maintenance as well). In addition, agricultural practices related to the maintenance of EFA were identified based on the FADN data for 2015. There were 4700 farms with EFA.

### 3.3. Data Sources and Methods: Case Studies of Polish Ecological Food Producers

The cases presenting the conditions and effects of eco-innovations in Polish agriculture covered five certified organic food producers, who were selected for the in-depth study. The main criterion of case sampling was the various expert opinions, which were based on their knowledge and professional experience. The agricultural advisers, members of organisations popularising organic food and the article's authors were inter alia among those experts. The cases analysed in the article were located in five different regions of Poland (NUTS-2 level): Lubelskie, Świętokrzyskie, Łódzkie, Mazowieckie and Podlaskie (Figure 1).

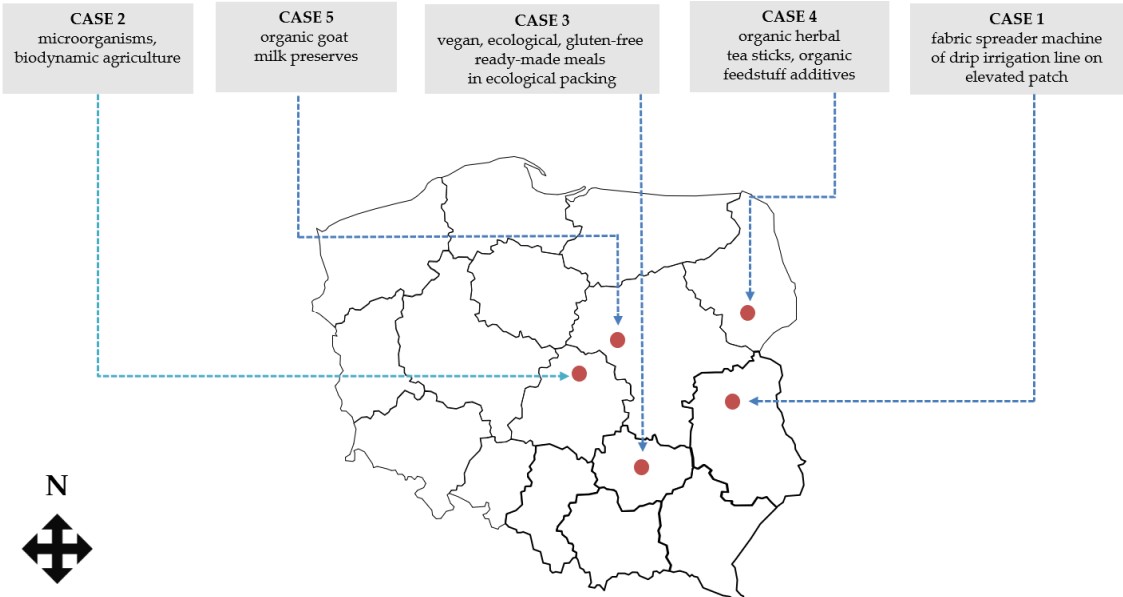

**Figure 1.** Location of case studies in regions of Poland and examples implemented of eco-innovations in agriculture. Source: own material.

The study used analysis of in-depth interviews, analysis of documents, and websites, as well as the direct observation (on-the-spot observation). The interviews with respondents were semi-structured and based on a script. Interviews and observations were carried out in June and July 2019. In all five cases studied, respondents were at the same time the owners or co-owners of these businesses. In the organic food producers studied included the following types of ecological innovation: product (good that is new or significantly improved), marketing (e.g., innovations concerning new ways of product selling or promoting), process (concerning the production process, new techniques and technologies) and organisational (method in workplace organisation) [36].

## 4. Results

### 4.1. Agriculture in Poland—The General Picture

Agriculture in Poland is very fragmented. In 2016, there were 1.4 million individual farms with at least 1 ha of agricultural land (Table 1) with the average surface 9 ha of agricultural land per farm. After accession of Poland to the EU in 2004, agriculture has changed significantly in terms of farm number and their production potential. There was observed decrease in farms' number almost 1/5 in comparison to 1.7 million farms in 2005. Those numbers indicated the withdrawal of many farmers from this economic activity. At the same time, human labour inputs in agriculture have been significantly reduced. Important factor was the changing agricultural production technology, resulting from farms' modernisation. The observed substitution of human labour on farms for objectified labour (costly investments or changes in the agricultural production technology by simplification, automation and mechanisation of this process) stemmed, to a significant extent, from the support for agriculture under rural development programmes (covering co-financing of building equipment and the purchase of agricultural equipment). Simultaneously, the area of agricultural land in good agricultural condition was around 13 million. The area in absolute terms increased by 121 thousand ha in the period, which was the result of the introduced commitments relating to the receipt of direct payments by maintaining land in good agricultural condition.

**Table 1.** Farms' numbers, production and economic potential of individual (Total) and organic farms (Org.) in Poland.

| No. | Specification | 2005 | | | 2016 | | | 2016/05 [1] | |
|---|---|---|---|---|---|---|---|---|---|
| | | Total | Org. | Org./Total [1] | Total | Org. | Org./Total [1] | Total | Org. |
| | Number of individual and organic farms | | | | | | | | |
| 1 | Farms' numbers (thous.) | 1723.9 | 3.0 | 0.2 | 1398.1 | 16.2 | 1.2 | −18.9 | 431.8 |
| 2 | UAA (thous. ha) [2] | 13,060.6 | 58.6 | 0.5 | 13,181.4 | 608.1 | 4.6 | 0.9 | 937.8 |
| 3 | Labour input (thous. AWU) [3] | 2035.2 | 5.6 | 0.3 | 1617.0 | 22.1 | 1.4 | −20.6 | 293.1 |
| 4 | Livestock (thous. LU) [4] | 6430.3 | 20.0 | 0.3 | 5923.5 | 116.9 | 2.0 | −7.9 | 483.7 |
| 5 | Livestock farms (thous.) | 1247.6 | 2.4 | 0.2 | 712.6 | 8.9 | 1.2 | −42.9 | 265.0 |
| 6 | SO (m. EUR) [5] | 20,824.1 | 70.3 | 0.3 | 21,824.3 | 817.3 | 3.7 | 4.8 | 1062.8 |
| 7 | SGM (thous. ESU) [6] | 9963.9 | 33.4 | 0.3 | 9283.4 | 405.6 | 4.4 | −6.8 | 1114.4 |
| | An average individual and organic farm | | | | | | | | |
| 8 | UAA (ha) | 7.58 | 19.3 | 154.7 | 9.43 | 37.7 | 299.5 | 24.4 | 95.2 |
| 9 | Labour input (AWU) | 1.18 | 1.9 | 56.5 | 1.16 | 1.4 | 18.1 | −2.0 | −26.1 |
| 10 | Livestock [7] (LU) | 5.15 | 8.3 | 60.2 | 8.31 | 13.2 | 58.9 | 61.3 | 59.9 |
| 11 | Stocking density (LU/ha) | 0.49 | 0.3 | −30.6 | 0.45 | 0.2 | −57.2 | −8.7 | −43.9 |
| 12 | SO (EUR thous.) | 12.08 | 23.2 | 91.6 | 15.61 | 50.6 | 224.3 | 29.2 | 118.7 |
| 13 | SGM (ESU) | 5.78 | 11.0 | 90.3 | 6.64 | 25.1 | 278.3 | 14.9 | 128.4 |

[1] In per cent. [2] Utilised Agricultural Area, UAA. [3] Annual Work Units, AWU, where 1 AWU is equivalent to labour inputs amounting to 2120 h a year [57]. [4] Livestock Units, LU, where 1 LU is a conventional unit of farm livestock with a mass of 500 kg. Conversion coefficients for livestock groups are presented in, e.g., [58]. [5] Standard Output, SO. [6] Standard Gross Margin, SGM expressed in European Size Units, ESU, where 1 ESU is equivalent to EUR 1200 of standard gross margins (SGM). SGM indicates the economic size of the farm, otherwise the productive potential of the farm. SGM on a particular crop or livestock is a standard (average of three years in a particular region) value of production obtained from one hectare or from one livestock less the standard direct costs necessary to produce [59]. [7] Livestock per an average livestock farm. Source: Own calculations based on aggregated Statistics Poland data from Farm Structure Surveys.

During the examined period, there were significant changes in the field of carried out agricultural production. Many farms resigned from livestock production—the number of farms with livestock decreased by 43% (from 1.3 million farms in 2005 to 713 thousand farms in 2016). It was tantamount to the growing population of non-livestock farms, which in previous years conducted livestock production as secondary agricultural activity. Livestock production requires significant labour inputs, the large involvement of farmer in daily on-farm duties and investments connected with building equipment and animal welfare. Those processes interacted with both the prices of agricultural means of production (industrial feed prices), as well as the sale prices of agricultural products. In addition, the requirements of the EU with regard to animal welfare required to take the investment activities, which constituted a significant financial charge for farms, in particular small units [60]. The outflow of labour force from agriculture and farms' transformation related to agricultural production simplification, contributed to resignation from labour-intensive and demanding livestock production.

The presented numbers indicated that in the last decade the process of farms' specialisation was observed; the number of specialised agricultural holdings significantly increased (mainly specialised in field crops), at the expense of those with mixed production (especially mixed livestock and crop production). The significant part of non-specialised farms relinquished from livestock production during the period, solely or mainly in favour of field crops [61].

### 4.2. Development of Organic Farming in Poland

Organic farming is one of the most important agricultural systems beneficial for the environment. During the period analysed (2005–2016)—after Poland's accession to the EU—there was a dynamic, more than five-fold increase in the number of organic farms (from 3 thousand farms to 16 thousand farms; Table 1). This was accompanied by changes in the production and economic potential of organic farming (observed in the surface of agricultural land, standard output result and the value of standard gross margin, which increased several times). The results showed that organic farmers obtained benefits from this economic activity.

Despite the relatively high dynamics of organic farming development, it is still a niche system. In 2005 and 2016, these farms used 0.5% and 4.6% of UAA, respectively, and their respective standard output was 0.3% and 3.7%.

Organic farms differ significantly from conventional farms (Table 1). On average, they are characterised by definitely greater production and economic potential (e.g., in 2016 the average farm area was 9 ha, while organic farm 38 ha; approximately standard output 16 thousand EUR and 51 thousand EUR). The differences between these farm groups have intensified over time. The larger area of organic farms compared to conventional farms has an economic reason—less value added per area unit—which preconditions the need to pursue more extensive farming and to seek external sources of financial support, including those mainly in the form of governmental programmes.

Comparing the changes in organic farms to those among all farms, it should be noted that in general these two groups followed in the same direction, i.e., they increased their production and economic potential, although the scale of these changes was more dynamic in the case of organic farms. The figures presented in Table 1 indicated the more efficient use of labour in organic farms. These changes are also a derivative of organisational changes related, in particular, to the decreasing stock density on organic farms, which is falling much faster than in conventional farms.

This evidences the fact that farms, especially organic ones, are simplifying agricultural production, that is expressed in resignation from agricultural production diversification (reduction of the conducted agricultural activities e.g., livestock production abandoning) and farms' specialisation increase. These changes contradict the idea of the functioning of organic farms, which should provide for a closed circulation of macronutrients within the farm as a result of combining the crop and livestock production. It is alarming that the proportion of organic farms with mixed (crop and animal production) has drastically decreased, simultaneously in favour of farms oriented on crop production [53]. Organic farms

increasingly target only crop production, both traditional—connected with arable farming—as well as orchards, while some of them use only permanent grassland—meadows and pastures.

### 4.3. Organisational Eco-Innovations in Agriculture at National Level—Implementation of the CAP Greening Mechanism in Poland

An important role in the implementation of changes in crop production should be attributed to administrative mechanisms. One of the most recent environmental solutions is greening. Taking into consideration the long history of the CAP, this instrument is an innovative ecological solution.

The average agricultural area of the analysed FADN farms obliged to implement greening was 44 ha. This group was dominated by larger farms (with an arable area of 15 ha or more), which had 92% of the total area of arable land, indicating the significance of their production organisation for nature protection. Larger farms are particularly important in the implementation of environmentally friendly practices, related to crop rotation and EFA maintenance [62]. Comparing 2015 and 2014, the production capacity of these farms did not change, which is a positive observation in the context of food security (approximately 227 thousand ha and 229 thousand ha of agricultural land; Table 2).

**Table 2.** Farms' numbers and land use [thousand ha] in Polish FADN farms obliged to greening implementation.

| No. | Specification | 2014 | 2015 | 2015/ 2014 | 2014 | 2015 | 2015/ 2014 | 2014 | 2015 | 2015/ 2014 |
|---|---|---|---|---|---|---|---|---|---|---|
| | | Total | | % | 10–15 ha | | % | ≥15 ha | | % |
| 1 | Farms' numbers | 5705 | 5705 | 100 | 1297 | 1297 | 100 | 4408 | 4408 | 100 |
| 2 | UAA | 251.7 | 253.5 | 101 | 25.0 | 24.8 | 99 | 226.7 | 228.7 | 101 |
| 3 | —Arable land | 225.4 | 227.9 | 101 | 19.5 | 19.3 | 99 | 205.9 | 208.6 | 101 |
| 4 | —Grassland | 25.3 | 24.6 | 97 | 5.2 | 5.2 | 100 | 20.1 | 19.5 | 97 |
| 5 | Cereal | 150.20 | 147.26 | 98 | 13.58 | 13.18 | 97 | 136.62 | 134.09 | 98 |
| 6 | Pulses for grain | 6.70 | 12.14 | 181 | 0.40 | 0.68 | 173 | 6.31 | 11.46 | 182 |
| 7 | Industrial | 39.03 | 37.95 | 97 | 1.20 | 1.15 | 96 | 37.82 | 36.80 | 97 |
| 8 | Potatoes | 4.01 | 3.98 | 99 | 0.62 | 0.55 | 88 | 3.39 | 3.43 | 101 |
| 9 | Fodder | 20.33 | 22.14 | 109 | 3.07 | 3.21 | 104 | 17.26 | 18.93 | 110 |
| 10 | —grasses | 2.89 | 3.45 | 119 | 0.51 | 0.61 | 121 | 2.38 | 2.83 | 119 |
| 11 | —pulses | 0.13 | 0.22 | 174 | 0.03 | 0.02 | 75 | 0.10 | 0.20 | 201 |
| 12 | —papilionaceous | 0.98 | 1.58 | 161 | 0.14 | 0.20 | 136 | 0.84 | 1.38 | 165 |
| 13 | Winter crops | 123.67 | 122.46 | 99 | 8.06 | 7.80 | 97 | 115.61 | 114.66 | 99 |
| 14 | Catch crops | 5.70 | 11.66 | 204 | 0.39 | 0.32 | 82 | 5.32 | 11.34 | 213 |

Source: Own calculations based on FADN data.

Meeting crop diversification and EFA requirements entails a specific pattern of cultivated crops. Incorporating winter and spring crops in crop rotation significantly facilitates the fulfilment of the crop diversification requirement. In both 2014 and 2015, the farms analysed used a significant part of their land for growing winter crops (accounting for more than half of arable land). The status quo as regards the area under winter crops was preserved, which should be seen as positive in the context of greening.

The cropping pattern in Polish FADN farms was dominated by cereals, with a 65% share, followed by industrial crops, which account for 17% of the total crop area. The share of remaining crops, including crops improving the soil structure, i.e., pulses and papilionaceous, is negligible and accounts for only a few per cent. Pulses and papilionaceous crops, both edible and fodder ones, are, however, a very important element of the cropping pattern and EFA requirement, which has beneficial effects on the amount of organic matter in soil, and consequently on soil productivity. Comparing 2015 and 2014, it should be noted that the area sown with pulses increased sharply, especially in larger farms (in total farms, the pulses sowings covered approximately 7 thousand ha and 12 thousand ha in 2014 and 2015). These changes occurred both in smaller farms, which chose pulses as a crop diversification element, and in larger ones, which were also obliged to maintain EFA (in the case of smaller farms their area covered approximately 0.4 thousand ha and 0.7 thousand ha in 2014 and 2015, while this

area in larger farms (≥15 ha), the area increased from 6 thousand ha to 11 thousand ha). The results reflect the impact of the legislation relating to greening on farmers decisions as regards the cultivation of soil-improving crops (Table 2).

Catch crops are another important EFA element. Their significance is due to their beneficial role in soil protection and improving soil structure. However, catch crops are a complementary element of the cropping pattern in Polish farms (in 2014, they made up only 2.5% of the cropping pattern in an average farm). It should be noted, however, that the area under catch crops increased rapidly in 2015 (from almost 6 thousand ha in 2014 to 12 thousand ha in 2015; Table 2). These changes occurred primarily in larger farms (approximately 5 thousand ha and 11 thousand ha), which proves the effectiveness of the greening in encouraging farmers to maintain EFA by using agricultural practices.

The legislation specified many different elements of EFA related to agriculture, forests and landscape. As indicated in Table 3, in 2015, the total weighed ecological focus area in FADN farms covered 15 thousand ha and 6.5% of arable land. These figures show that the farms analysed fully complied with the requirement to maintain EFAs (taking into account the result for the entire farm groups analysed and institutional obligation 5% of arable land under EFA). It is worth underlining that farmers did not diversify EFAs—at farm level, 94% of farms selected one or two EFA types from the list of 14 different EFA elements. Farmers concentrated on suitable crop production, adjusted to environmental requirements (87% of the weighted ecological area was used for stubble catch crops and the cultivation of nitrogen-fixing plants; Table 3).

**Table 3.** The main EFA * elements in Polish FADN farms in 2015 (number and percentage of farms that selected the most popular type of EFA and the surface of EFA).

| Elements | Farms | | Surface of EFA in ha | | Surface of EFA in % | |
|---|---|---|---|---|---|---|
| | Number | % | under Conversion | Weighed | under Conversion | Weighed |
| Stubble catch crops | 2707 | 57.1 | 16,749 | 5025 | 54.2 | 34.2 |
| Nitrogen-fixing crops | 2229 | 47.0 | 11,173 | 7821 | 36.1 | 53.2 |
| Winter catch crops | 275 | 5.8 | 1610 | 483 | 5.2 | 3.3 |
| Fallow land | 228 | 4.8 | 804 | 804 | 2.6 | 5.5 |
| EFA in total | 4744 | x | 30,910 | 14,699 | 100 | 100 |

* EFA elements: (EFA1) fallow land; (EFA2) hedges; (EFA3) single trees; (EFA4) trees in line; (EFA5) trees in group; (EFA6) field margins; (EFA7) ponds; (EFA8) ditches; (EFA9) buffer strips; (EFA10) land strips without production along forest; (EFA11) land strips qualified for the payment, located along forest edges; (EFA12) short-rotation coppice; (EFA13) afforested areas; (EFA14a) stubble catch crops; (EFA14b) winter catch crops; (EFA14c) under sown grasses; (EFA15) nitrogen-fixing crops. Source: Own calculations based on FADN data.

Summing up, in the first year of greening implementation, the FADN farms obliged to greening met the legal requirements, which justifies its effectiveness in the light of the established agricultural practices [59]. According to research on FADN data, innovative administrative solutions based on conditional support for agriculture guarantee the popularisation of the desired practices. The research results indicated the importance of agricultural production organisation in the context of protection of the environment.

*4.4. Drivers Behind Eco-Innovations and Their Effects at the Local Level—Results of Case Studies*

4.4.1. Human Capital, Labour Input and a Pro-Active Attitude towards Farming

A crucial element in the successful implementation of eco-innovations in the cases analysed should be attributed to the characteristics and attitudes of the respondents (Table 4). Decisions on introducing new products, services and organisational improvements were preceded by organic farmers obtaining specialist and appropriate knowledge. In their cases, valuable knowledge was acquired through individual learning and learning by doing, as well as from the experts (e.g., agricultural advisers and agri-business companies' employees). Unique information was also passed on to the eco-innovators by

their family members, members of the local community (e.g., knowledge and skills about construction and repair of agricultural machinery provided by the father (Case 1) or the specialist information on organic vegetable growing provided by a local agricultural adviser (Case 3)).

In the process of extending competences and business projects the selected cases of eco-innovators were characterised by specific personal characteristics and a pro-active attitude towards development of food production. In particular, they showed persistency, diligence and entrepreneurship (in terms of economic calculations and good organisational skills), openness to change, willingness to take risk and ease of making business contacts and establishing relationships with people. For example, Case 2 was one of the first vegetable growers running their orchard according to organic and biodynamic agriculture principles. The eco-innovator obtained the necessary knowledge as a member of a farm network—Ecological Folk High School training courses and during the study tours. In turn, Case 5 acquired the knowledge about goat breeding and goat's milk processing thanks to specialist foreign literature from a local goat breeder and scientists from the Warsaw University of Life Sciences, as well as by participating in specialist cheese-making courses.

### 4.4.2. Organic Production Issues

In all cases of eco-innovators analysed, a decision to switch to organic food production was declared as a turning point for running a business. It should be emphasised that running this kind of agricultural production system in the neighbourhood or in eco-innovators' local area was rarely adopted. The conversion to organic production gave an impetus to experimentation, establishing market relationships and acquiring valuable knowledge and helped to develop the business. In Cases 1, 3 and 5, setting up certified organic food production was followed by starting cooperation with bigger processors and retailers (intermediaries in the food trade, food stores, markets). In Cases 2 and 4, the certificate of organic production contributed to raising the profile of their products on the market (increased demand for organic apples and herbs, respectively) and gaining favourable sales prices.

### 4.4.3. Importance of the Background and Revalorisation of Local Resources: Farm Succession and New Use of Natural and Physical Capital

The cases of eco-innovations in agriculture presented showed the ambiguous role of the strength of ties of innovators with tradition, locally or family-embedded ways of food producing or with the place of origin. On the one hand, behind successful implementation of ecological innovation were the natural resources and tangible assets (inherited or purchased land, agriculture machinery and tools—Cases 1, 2, 3 and 5), as well as intangible assets (e.g., interests, hobbies, skills—Cases 1 and 4) passed on by family members or people from the local community. On the other hand, the resources such as agricultural land, buildings and machineries, as well as knowledge and interests in food production, were used by eco-innovators in a different way. In Case 3, an herbal hobby taken over from grandmother was a trigger for setting up the business. Moreover, four of the five cases analysed were based on locally or family-embedded economic activities and goods (for Cases 3 and 4, the brand name and type of food produced strongly related to the regional tradition and, in Case 2, the farm is located in the most important Polish fruit-farming area).

In the examples of agricultural eco-innovations analysed, a conducive factor for generating the new products and organisational solutions was the economic potential of the farm, i.e., relatively large area of land as for the conditions of Polish agriculture (especially in Case 1, 3 and 4—30, 348 and 60 ha of UAA, respectively) (Table 4), as well as the physical capital acquired (machinery, buildings and tools—Cases 1–4). A critical element that stimulated innovation was also a good financial situation, which created a favourable background for experimentation and risk taking (Cases 1–4).

**Table 4.** Characteristics of selected food producers and eco-innovations.

| Specification | Case 1 | Case 2 | Case 3 | Case 4 | Case 5 |
|---|---|---|---|---|---|
| Region in Poland | Lubelskie | Łódzkie | Świętokrzyskie | Podlaskie | Mazowieckie |
| Information about manager | agricultural education: vocational, 10 years of experience in farming | economist, specialist and adviser in ecological and biodynamic agriculture, 30 years in fruit farming | higher education in environmental engineering, 19 years of experience in farming | PhD in agricultural sciences, 29 years in herbiculture | trained in milk and cheese-making, 23 years of experience in goat's milk processing |
| Size of farm * | 30 | 19 | 348 | 60 | 0 ** |
| Employment *** | 2 | 4 | 150 | 300 | 6 |
| Type of agri-food production | horticulture | permanent crops | horticulture, vegetable processing | permanent crops, horticulture | agri-food processing |
| Products | strawberries, raspberries | apples, sweet cherries | tomatoes, courgettes, carrots, cucumbers | herbs, herbal teas, bio-cosmetics, dietary supplements | goat's milk, goat's cheese |
| Type of innovation | process | organisational | organisational, product, marketing | product, organisational | product |
| Innovation introduced | fabric spreader machine of drip irrigation line on elevated patch | beneficial microorganisms, biodynamic, regenerational agriculture, orchardists' association | vegan, ecological, gluten-free ready-made meals in ecological packing | organic herbal tea sticks, organic feedstuff additives, management of herb pickers' work | organic goat milk preserves, organic goat's milk cheese balls |
| Drivers of innovation | involvement, high mechanic's skills, early farm succession, demand on local market, credit | early farm succession, social capital, human capital, switching to organic production, experiments | early farm succession, switching to organic production, skills, involvement, social capital, local demand | hobby, family tradition, specialist knowledge, switching to organic production, skills, demand, local resources | hobby, development of knowledge and skills, specialist training |
| Effects of innovation | cost limitation, yield increase | multiplying bio-diversity, market stand-out | profit increase; waste limiting, business development and diversification | herb protection and development, business development and diversification | filling a market niche, healthy food |

* Hectares of UAA. ** The farm was run only at the initial stage of business development for the purpose of goat breeding. Then, the manager leased the whole land, i.e., liquidated the agricultural holding, and concentrated exclusively on milk processing activity. *** The number of persons employed (full and part-time).

### 4.4.4. Types of Eco-Innovations and Their Effects

The cases of organic food producers presented in the article cover various examples of innovation, including product marketing, process and organisation (new goods, way of selling, functioning and technologies) (Table 4). All food producers analysed were eco-friendly, e.g., activated natural production mechanisms, created positive external effects by producing high quality and healthy food, promoted healthy lifestyle and guaranteed high livestock welfare standards (the latter referred to Case 5). For instance, the herb pickers employed in Case 4 work within the organisational system, limiting damage to forest flora and promoting practices favourable for growth and spreading plants. The natural products with specially selected compositions of beneficial microorganisms (probiotic substances), applied in the orchard in Case 2, improved soil fertility, supported the growth and health of apple trees and at the same time increased the biodiversity. In turn, food products offered by Case 3 have health-promoting properties (no genetic modifications) and were sold in organic and safe packaging.

Apart from positive environmental effects, the organisational solutions implemented in food production brought economic benefits for the respondents. These included acquiring new clients and increased economic margins, raised market profile or obtaining external financial support. For instance, in Case 5, the organic cheese from goat's milk was a product innovation responding to a demand from a specific market segment, i.e., clients suffering from an allergy and consumers who appreciate healthy, tasty and niche food. A new and enhanced fabric-spreader machine used in Case 1 enabled the development of strawberry plantation in wet areas while limiting costs and providing considerable growth in yields. The novel and unconventional use of herbs in Case 4 made implementation of subsequent business projects supported from public funds easier. These projects included a modern laboratory for examination of harmful pesticide residues and other pollutants used in agriculture, the production of organic feedstuff additives for animals, the botanical gardens, an educational centre for various age groups and the manufacture of bio-cosmetics. It can be stated that in the cases analysed a process of accumulation of the same or different types of innovation took place. Specifically, the creation of a new product was followed by developing further unique commodities (organic herbal tea sticks containing specially selected herbs were inspired by offered earlier organic herbal products (Case 4) or the experimentation with pumpkin processing that paved the way to making the recipes of organic preserves and ready-made meals (Case 3)).

The cases observed exemplified a number of organisational innovations. These covered creating a unique and often bottom-up business model between food producers (bringing together organic farmers and entrepreneurs within associations), as well as between the producers and customers. For instance, the innovation reported in Case 2 concerned different forms of collaboration between organic fruit-growers (association and informal group), enabling the implementation of common business strategy on the market with the use of new technologies (social media, software and ICT). Ecological innovations covered by the study reflected not only sustainable use of natural resources but also running a locally unusual human-resources policy based on a trust between superiors and employees, a high level of salaries and the recruitment of employees above all from a local area or neighbourhood. Sustainable human resource management referred especially to Cases 3 and 4 due to the considerable number of employees they hired (150 and 300 employees, respectively; Table 4). The creation of an own brand name by Case 3 could be considered as product innovation. This step in food production and processing made the farm and company independent of intermediaries and locally constituted a market novelty—offering products in direct sales via an online store.

## 5. Discussion

The conventional approach to eco-innovations in agriculture presumes the primacy of an economic interest over an ecological one, because it does not challenge the paradigm of growth, which contributes to exploitation of natural resources and to climate change [63]. In addition, the traditional depiction of ecological innovations focuses on technologies in which eco-friendliness is usually based on an increase

in production efficiency [24,63]. However, it is argued that their potential users do not generally like the implementation of new products and services with the high costs and uncertainty. Furthermore, modern technologies cover specific solutions targeted at economies of scale and narrow product specialisation. Hence, the majority of conventional ecological innovations are mainly not applicable to smaller units, farms, organic and biodynamic production systems [30].

The limitations of the conventional approach to eco-innovations in agriculture encourage the search for an alternative. One such alternative may be a perspective relating to the concept of sustainable development. In food production, this approach could generally be defined as all activities aimed at improving the relationships between the farm or business entity and the natural environment, which at the same time brings social and economic benefits [39]. Nevertheless, this understanding of the issue is dependent on the temporal and spatial context. Thus, it is to reflect a wide spectrum of products, services and solutions. Ecological innovations fitting neatly into the sustainable development concept should cover the phenomena appropriate for food producers to implement new activities both for the farm (cutting-edge for themselves) and at local and regional levels.

The examples of product, process, marketing and organisational solutions illustrated in the paper related to a sustainable approach to eco-innovations in agriculture. The analysis of organic food producers in this study showed that the successful implementation of ecological innovations was conditioned by several individual, social, economic and institutional factors affecting different levels (microeconomic, local and regional) (Figure 2). Among these factors, the respondents' individual characteristics associated with knowledge, attitudes towards farming and the social, human and physical capital passed on by family members should be highlighted. To create eco-innovations presented in the study, the favourable influence had the local natural and cultural context reflected in the prevalent type of farming (crops) and the historically embedded activities, services and habits. Generating new products, processes and organisational solutions was also defined by the market conditions in terms of demand for the eco-products offered, as well as by the possibilities of using business expertise of representatives of various branches and disciplines. In the process of achieving the ecological innovations an institutional factor played an important role as well. This concerned both the system of organic production certification, public programmes supporting research and development (R&D) activities, financial measures available within the EU CAP and bank loans. What should be stressed is the fact that, in the emergence of new products, processes and organisational solutions, the above-mentioned components did not play a significant and pro-innovative role separately. Rather they constituted a complex of reasons that affected the enterprises analysed with different degrees of intensity and in different moments of development.

As indicated in the case studies, agricultural policy is an important and effective stimulant in creating eco-innovation at farm level. Positive transformations in agricultural sector were also evident, both in the case of relatively new mechanisms such as greening and in the instruments that initiated agri-environmental direction of the CAP, e.g., organic-farming support.

Based on the example of greening implementation presented in the paper, organisational eco-innovation may be administratively motivated. Greening is a major innovation brought in under the 2013 CAP reform, making the system of direct payments more environmentally friendly. Mandatory green standards connected with direct payments in the first pillar of the CAP were defined as a novel approach [64]. "It was designed to reward farmers for applied as part of the agri-environment and climate measure having a positive impact on the environment which would otherwise not be rewarded by the market" [65]. The introduction of the new greening measures was a significant but controversial aspect of the CAP reform [66]. Research based on modelling echoed this argument, indicating that, in the present form, the environmental impacts are rather limited and will not contribute much to improving the CAP provision of public goods [67,68]. There are indications that the CAP greening needs to be redefined and regionalised to ensure the transition towards "greener" agriculture [69]. However, it was important political step promoting a better environmental performance by the EU farming sector [70].

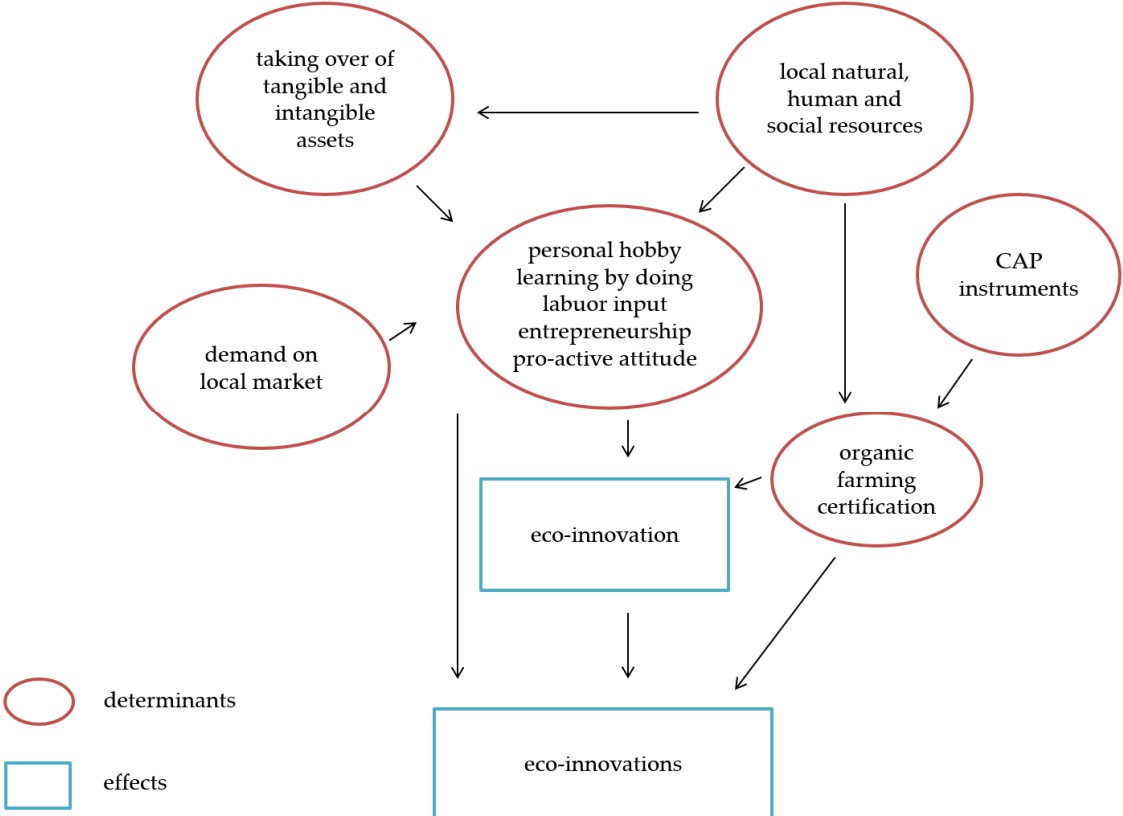

**Figure 2.** Scheme for achieving eco-innovation based on the case studies. Source: own elaboration.

The study showed that Polish FADN farms met the mandatory greening requirements and indicated that innovative administrative solutions based on conditional support for agriculture guarantee the popularisation of the desired environmentally and climate-friendly practices. The greening mechanism was effective in the light of the specified requirements for agricultural production. In the first year of its implementation, these requirements did not adversely impact the production and economic outcomes, because the area allocated to the ecological focus areas amounted to just a few per cent of the area in use and the crop-diversification criteria did not force any significant organisational change in crop production. The environmentally friendly organisation of Polish farms before the introduction of the greening requirement allowed them to adjust smoothly in 2015. Maintaining the status quo on farms (as regards winter crops) or implementation of the desired organisational changes in crop production is the quintessence of the measures related to meeting the greening requirements. In the context of soil-improving crops, other measures such as agri-environmental programmes and direct support have encouraged farmers to cultivate crops with an environmental "symbiosis".

An important part of agricultural sector is formed by organic farming. This is growing rapidly (as indicated by the multiple growth of the number of organic farms and their production and economic potential) although its range is still niche, as the model of conventional (industrial) farming is dominant in developed countries [16]. The major determinant of this process was the subsidising of organic farms from rural development funds. This is undoubtedly an example of organisational eco-innovation in agriculture. In view of the promising forecasts of the demand for organic food and the planned subsidising of organic production in the next EU budget perspective, we can expect the further development of organic farms.

However, substantive criteria of the organic production system should be subject to discussion, market processes force specialisation and the simplification of agricultural production at farm level. Assuming that the current legislation is maintained, the further simplification of the agricultural production in organic farms is highly probable, which may contradict the need to implement practices

consistent with the idea of sustainable development. Livestock production elimination is determined by economic factors and animal production requirements within the scope of field crop structure, and also—or primarily—by the need, dictated by the market, to narrow the specialisation of agricultural production (large and uniform batches of goods). The principles of sustainable development indicate the importance of a closed circuit of organic matter and nutrients within a farm. Organic farms, without animal production and natural fertilisers (mineral too, because of legal requirements) can face organisational problems in balancing important soil ingredients, because animal breeding predetermines the proper functioning of the agricultural ecosystem, which is the guiding principle of organic farming [71]. The purchase of natural fertilisers is a partial solution, but it is not a popular practice on the market. An important reason for the retreat from livestock production in organic farms was the underdeveloped processing sector, in particular meat processing [72]. Agricultural eco-innovation should take account of the development of livestock production, which requires on the one hand the verification of current legal requirements and on the other hand support for the development of food-chain links. It is desirable to implement various environmental innovations, not only in agriculture but in the processing sector as well.

The quantitative and qualitative analysis of empirical data carried out on the functioning of organic food producers, which was based on the big data bases and case studies, gave an opportunity to consider ecological innovation in agriculture as a complex, multidimensional and ambiguous phenomenon. Firstly, the understanding of eco-innovation varies. Recognising a particular product, service or solution in a farm's activity as eco-innovative may not be straightforward. Secondly, the capture of novelty or unconventionality of a given agricultural practice brings the need to assess the effects of its implementation. It should be emphasised that the consequences of eco-innovation simultaneously affect many areas of life. According to the concept of sustainable development, their specificity and significance should be considered within three orders: environmental, social and economic.

It is undoubtedly appropriate to continue research on eco-innovation in agriculture. Research on the importance of different types of eco-innovation, including the environmental and economic costs and benefits associated with their implementation, is of immeasurable importance. In view of the increasing environmental and climate problems, any consideration improving the current state of affairs or eliminating the negative processes is justified. Different supporting instruments (including policy measures) and these criteria of access to them will play an increasingly important role in eco-innovation. Taking into consideration the change in consumer preferences related to their environmental awareness and thus the change in demand preferences, the supply of products, including agricultural ones, will have to adjust to the "new" requirements. These issues still require scientific verification and an interdisciplinary approach. This paper presents only examples of agri-eco-innovation approaches, as well as an empirical illustration in this regard indicating the scale of changes in Polish agriculture in this area.

## 6. Conclusions

The aim of the study was to provide the examples of eco-innovations in agriculture relating to the concept of sustainable development and indication of their conditions. Basing on the Polish FADN and Statistics Poland data, the study analysed the agricultural policy instruments addressed to the farms that were considered as the examples of organisational innovations with relatively low popularity in the food market (organic farming system) and at the early stage of their implementation (greening mechanism within the EU CAP introduced in 2015). Comparative analyses of data from the Polish farms showed that an EU policy schemes contributed to realisation of environmental aims concerning the increase in ecological friendliness of agricultural production and the counteraction of a climate change.

Firstly, during 2005–2016, the certification system of organic production, as well as policy support contributed to the development of organic farms in Poland in terms of the growth in both the share of this type of agricultural holdings (from 0.5% to 4.6%) and the total cultivated agricultural land (from

0.3% to 3.7%). Despite the relatively high dynamics of organic farming development, it is still a niche system. In addition, organic farms are significantly different from conventional farms in terms of their potential and production orientation—they are considerably larger and economically stronger (on average). In the analysed period, the advantage of organic farms was growing. Comparing organic and conventional farms, changes in agricultural production simplification and specialisation concerning organic farms are more intensive.

Secondly, the greening mechanism had a positive impact on innovativeness of Polish farms by favouring agricultural practices that are more sustainable from the environmental point of view. During 2014–2015, in the agricultural holdings, the area sown with pulses and papilionaceous, i.e., crops improving soil structure and protecting soils, rose by 174% and 161%, respectively. According to research, innovative administrative solutions based on conditional support for agriculture guarantee the popularisation of the desired practices. The research results indicated the importance of agricultural production organisation in the context of protection of the environment.

In the study, the process of emergence of eco-innovation in agriculture was subjected to in-depth analysis. Analysis of the case studies of selected ecological food producers in Poland documented that using appropriate family and local tangible, as well as intangible assets in a specific way, contributed to creation of innovative products, processes and marketing and organisational solutions that were pro-ecological and economically beneficial. These examples illustrated that stepping into ecological food production with new ideas, social relations, material resources and often non-agricultural background constitute the important determinants of business development and cause the positive externalities for the natural environment and local societies.

**Author Contributions:** Conceptualisation, M.D. and W.W.; methodology, M.D. and W.W.; software, M.D. and W.W.; validation, M.D. and W.W.; formal analysis, M.D. and W.W.; investigation, M.D. and W.W.; resources, M.D. and W.W.; data curation, W.W.; writing—original draft preparation, M.D. and W.W.; writing—review and editing, M.D. and W.W.; visualisation, M.D. and W.W.; supervision, M.D. and W.W.; project administration, M.D.; and funding acquisition, M.D. All authors have read and agreed to the published version of the manuscript.

**Funding:** This research was funded by Agricultural Advisory Centre in Brwinów, Poland within National Rural Network Competition No. 3/2019, agreement No. KSOW/3/2019/017 from 14.06.2019. Project titled "How do eco-innovations in farms emerge? Analysis and examples" co-financed from the EU budget within Scheme II of Technical Assistance "National Rural Network" of the Polish Rural Development Programme for 2014–2020.

**Acknowledgments:** We are grateful to our research participants for the interviews, shared experiences and donating their valuable time to our study. We would also like to thank our colleagues Marcin Żekało and Konrad Prandecki, who participated in the research project as a co-investigator. Furthermore, we thank the reviewers for their detail reviews of our manuscript and very valuable suggestions.

**Conflicts of Interest:** The authors declare no conflict of interest.

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
