# Peer review of "On the Way to Eco-Innovations in Agriculture: Concepts, Implementation and Effects at National and Local Level. The Case of Poland"

_sustainability, doi:10.3390/su12124839_

Round 1
Reviewer 1 Report
The topic is very important and very interesting, so it should be presented in a way that will allow the use of ecological innovations in further development. Now the text is incoherent and rushed, and can’t be published in this form. Please make deep revision.
Figure 2 – some text (words) at the picture is cut. What is a difference between eco-innovation and eco-innovations (except the using the plural form) – please make this part of scheme more understandab le. What is a meaning of using different pattern of elements (square, oval). Please indicate if the difference between pattern of elements of the scheme are important, if not, please use the same style.
Please write full names of abbreviation when you use first time: e.g CAP, UAA, R&D
In figure 1, unify the writing of farm words. Sometimes it is a capital letter, sometimes a lowercase letter. In addition, in the text there is a different terminology (lines 235-238) Kozak’s farm etc than in the figure (horticultural farm, Swietokrzyska company not Swietokrzyska farmetc), but for Capra Campinos and Dary Natury is the same.
Table 4 – please write the size of the farm in the case 5. If there is n/a it seems very strange, because Authors made face to face interview, so why there is n/a?
Table 1 and 2 – please use the same style of numbers. Sometimes authors use comma for indicate thousands, sometimes they use space between numbers. Sometimes there is comma sometimes there is dot. Please explain abbreviation of LU, ESU
Table 3 - what is AWU? Please use Org or Org. (with or without a dot). Please make tables and drawings more carefully. Please use EUR or euro (No 6 and No 12). please change the headings in the table because it is unambiguous. e.g. the number of farms but in the table one of the position shows in euros. More precision in writing text, numbers, and abbreviations will be added value.
Table 4 - The authors did not use the information from the table (size and number of employees) in their analyzes. If not, why did they show this information?
The authors wrote that they conducted quantitative research (statistics). Unfortunately, the article has no statistics. Dividing one number by another is not a statistic. Please make statistical analysis in the article. There is Pearson correlation, chi square test and others.
Please provide newer data in the tables from the Statistical Office. Maybe 2019 but certainly 2018. When writing the article in 2020, showing data from 2014/2015 and 2005/2016 is incomprehensible. In addition, these data are not source (statistical year) but are given after another publication. First of all: why 2014 and 2015 and next 2005 and 2016. Is any reason for that? Now it looks like: we had those data so we use those, next we had another data so we used another data. In addition, the authors realized the qualitative research in 2019 so there is no justification for showing tables with data from 2014 and 2015 and compare with situation from 2019. Please update the data.
Please harmonize the innovation aspect throughout the document. Here I will give only an example: Once the authors mention two types: product and organizational (line 72). Other times, product, service, marketing and organizational (line 400) and in table 4 organizational, product and marketing. Please use more current international literature to show that the research is prepared for the article in 2020.
What does it mean in-deep perspective as a second type of the research? Authors wrote that they used both: quantitative and in–depth perspective. Maybe should be quantitative and qualitative (including in-depth interviews)? Now there is misunderstanding.
What is the main conclusion from the research. Please indicate more precisely, kind of highlights.
Did you receive permission from respondents to use their names and names of companies?
Author Response
Dear Reviewer,
Please see the attachment.
Best regards,
Authors

Reviewer 2 Report
The paper describes the effects of European Common Agriculture Policy on Polish farms. Effects were studied with three different analysis based on polish agricultural holdings dataset.
The study well describes the Polish agricultural sector and the eco-innovators key factors. By the way there are some parts that should be implemented.
As general comments I suggest to the authors to add more references for enviromntal benefits provided by organic farming.
Specific comments:
Introduction
Some English issues needs to be fixed. Please define CAP in a better way and summarize the importance of European cohesion policy (1-2 sentences)
Line 48: Sentence is note clear, please reformulates it
Line 54: Sentence is note clear, please reformulates it
Line 57: Please declare the abbreviation the first time you use it
Line 62: The positive contribution of organic farming to environment is not always true. The paper cited is a report from a German Green Party foundation. Please reformulate or remove it
Line 92: Industrial agriculture is a misleading term. Please describe what you mean
Line 189-190: Is this an author’s opinion? It needs reference
Methodology
Line 227: Which is the average farm size in Poland? 1 ha of land for agricultural holdings is quite low for arable crops while is suitable for vegetable or specialized crops
Figure 1: Maps must have scale and north indication
Results
Line 314: Not appropriate for results. By the way it needs reference
Line 389: This detail is not suitable for scientific articles
Line 416: New technologies as eco-innovators seems to lead the sustainable agriculture conversion. Authors should consider to add a specific paragraph about new machines and ICT as sustainability carrier
Author Response

(The authors gave the same response as above.)

Reviewer 3 Report
Please, see the attach with the comments and suggestion.

Author Response

(The authors gave the same response as above.)

Round 2
Reviewer 1 Report
Thank you for your explanations and changes.
One thing more: Please put ** under the table 4 to explain 0**
Author Response
Dear Reviewer
Thank you very much for your comment.
The footnote under the table 4 was corrected.
Best regards
Authors

Reviewer 3 Report
I guess the authors have made the corrections I had suggested.
Author Response
Dear Reviewer
Thank you very much for checking all of the corrections we have made in the manuscript and verification of our answers to your comments.
Best regards
Authors
